# The Effect of Maternal Trait Mindfulness on Preschoolers’ Social Competence: The Chain-Mediating Role of Maternal Self-Control and Problematic Social Media Use

**DOI:** 10.3390/bs13100805

**Published:** 2023-09-27

**Authors:** Jinxia Han, Shuo Feng, Ziming Wang, Jingyu He, Hehong Quan, Chun Li

**Affiliations:** 1Department of Education, Qufu Normal University, Qufu 273165, China; hanjinxia@qfnu.edu.cn; 2Department of Preschool Education, Qingdao University, Qingdao 266000, China; fengshuo@qdu.edu.cn (S.F.); wangziming@qdu.ecu.cn (Z.W.); hejingyu@qdu.edu.cn (J.H.); 3Graduate School of Humanities and Social Sciences, Hiroshima University, Hiroshima 7398511, Japan

**Keywords:** trait mindfulness, social competence, self-control, problematic social media use

## Abstract

The impact of maternal trait mindfulness on the development of preschoolers’ social competence is receiving increasing attention from researchers. However, the mediating mechanisms that link maternal mindfulness to preschoolers’ social competence are still not well understood. This study examined the mediating effect of maternal self-control and problematic social media use on the association between maternal trait mindfulness and preschoolers’ social competence. We administered 407 mothers of preschoolers in China a questionnaire to assess their trait mindfulness, self-control, problematic social media use, and the degree of social competence of their children. After controlling for demographic variables, the results showed that (1) Maternal trait mindfulness was positively related to preschoolers’ social competence; (2) Maternal self-control and problematic social media use independently mediated the relationship between maternal trait mindfulness and preschoolers’ social competence; and (3) Maternal self-control and problematic social media use play a chain-mediating role between maternal trait mindfulness and preschoolers’ social competence. These findings have enhanced our understanding of how maternal trait mindfulness influences preschoolers’ social competence and holds important implications for interventions aimed at enhancing preschoolers’ social competence.

## 1. Introduction

One of the most crucial skills for the social development of preschoolers is social competence. It describes the capacity to flexibly deploy behavioral, cognitive, and emotional resources to accomplish social goals in a specific situation [1]. It encompasses a range of fundamental abilities required for interpersonal and social interactions, including emotional comprehension and control, peer engagement, and adherence to social norms [2]. The development of social competence in early childhood is crucial for subsequent emotional, cognitive, and behavioral adjustment. Strong social competence helps children create and sustain strong interpersonal interactions [3,4].

Mothers are the major nurturers and important people in their children’s development [5], and their characteristics (such as trait mindfulness) can have a subtle influence on their children’s social competence [6]. In addition, with the rise in popularity of social media, maternal parenting styles are inevitably influenced by social media. However, due to the “fragmented” use of cell phones [7], it is challenging for mothers to exercise self-control over their frequent and automatic use of social media, even when they are with their children [7]. According to research, excessive social media use can hurt a person’s physical, mental, and interpersonal health [8]. Additionally, social displacement theory contends that maternal use of social media can replace or lessen meaningful parent–child interactions in the real world [9], which in turn affects how socially competent children become.

This study investigated the impact of maternal trait mindfulness on preschoolers’ social competence within the context of social media and looked at the function of maternal self-control and problematic social media use in mediating the chain in this process.

### 1.1. The Influence of Maternal Trait Mind on Preschoolers’ Social Competence

Trait mindfulness refers to a person’s capacity or propensity to practice mindfulness in daily life, i.e., to adopt an accepting and nonjudgmental attitude to concentrate on what is thought and felt in the present moment. This capacity or tendency is shared by all people, albeit it varies from person to person [10,11]. In parenting, trait mindfulness extends from personal introspection to interpersonal interactions between mother and child [12]. Mothers with higher levels of mindfulness tend to listen and respond to their children’s needs with full attention and acceptance of their own and their children’s emotions and thoughts, resulting in less impulsive behavior and more inclusive parent–child communication. Research has shown that mothers’ positive attention and expression of emotion to their children can predict their popularity among peers and the development of social skills [13]. Mothers with low levels of mindfulness tend to be insensitive to their children’s needs, and they often fail to effectively regulate their negative emotions in parenting and tend to criticize and blame their children [14], which greatly reduces the chances of preschoolers gaining social competence from positive interpersonal relationships [15]. This results in poor emotional regulation and more peer interaction problems [16], which in the long run can hinder the development of social skills. Based on this, this study suggests that maternal trait mindfulness positively predicts preschoolers’ social competence.

### 1.2. The Relationship between Maternal Trait Mindfulness, Self-Control, and Preschoolers’ Social Competence

According to research, self-control and trait mindfulness have a very strong and beneficial relationship [17]. Self-control refers to a person’s capacity to regulate their feelings, stifle undesirable or inappropriate behavioral inclinations in their minds, and refrain from acting following those intentions [18]. Effective self-control behaviors depend on the resources that are accessible, and the more adequate the resources, the better the individual’s self-control performance [19]. If a person’s self-control reserves are exhausted, this might impede socially adapted behavior and make them ineffective [20], such as when they have trouble controlling their emotional emotions [21] or act impulsively [22]. A maternal lack of self-control in family relationships may have an adverse effect on parent-child interactions through bad emotional and impulsive behaviors, which affect preschoolers’ social competence. According to research, mindfulness can improve personal self-control by reducing the resources that are depleted by distraction [23]. Rowland et al.’s intervention study from 2019 also discovered that even brief mindfulness exercises were successful in raising people’s levels of self-control [24]. Based on this, this study posits that a mediating factor for maternal trait mindfulness to affect early children’s social competence is self-control.

### 1.3. The Relationship between Maternal Trait Mindfulness, Problematic Social Media Use, and Preschoolers’ Social Competence

A growing number of women are using social media frequently and unconsciously while spending time with their children as a result of the popularity of social media. Undoubtedly, this is a worrying phenomenon. Li et al. (2022) found that parental screen addiction affects young children’s screen addiction through both direct and indirect paths. Further research has found that women are more likely than men to engage in harmful social media habits that affect their own psychological and physical well-being, as well as their ability to interact with others [25,26,27,28]. When moms’ requirements for social media use collide with their responsibilities as parents, they are more likely to experience negative feelings like anxiety and despair [8,29], which can affect how they interact with their children [30]. However, mothers with higher levels of mindfulness are acutely aware of their own emotional and behavioral issues and make timely self-adjustments to reduce impulsive reactions and avoid the negative effects of adverse parent–child interaction processes on preschoolers’ social competence [12,31,32]. Research has shown that maternal emotional expressions can affect preschoolers’ social competence, but mothers with higher levels of trait mindfulness are also more aware of their own emotional and behavioral problems. Based on the reality of social network context, this study suggests that maternal problematic social media use may serve as another mediating variable in the influence of maternal trait mindfulness on preschoolers’ social competence.

According to attachment theory, maternal psychology and conduct have a significant influence on how well their children develop social competence. As important nurturers, maternal trait mindfulness may have an impact on preschoolers’ social competence through their level of self-control and performance of problematic social media use. Research has shown that self-control negatively predicts problematic social media use behaviors [33]. In the context of the family, problematic social media use by mothers can also be viewed as a failure of self-control, as excessive social media use can significantly reduce the efficiency of face-to-face parent–child interactions and have a noticeable impact on the development of children’s social competence. Based on these findings, the influence of maternal trait mindfulness on preschoolers’ social competence is chain mediated by maternal self-control and problematic social media use.

In light of the statements above, this study presents the following hypotheses: 

**H1.** 
*Maternal trait mindfulness is positively correlated with preschoolers’ social competence.*


**H2.** 
*Self-control mediates the influence of maternal trait mindfulness on preschoolers’ social competence.*


**H3.** 
*Problematic maternal social media use is another mediating variable in the influence of maternal trait mindfulness on preschoolers’ social competence.*


**H4.** 
*Both maternal self-control and problematic social media use act as sequential mediators in the relationship between maternal trait mindfulness and preschoolers’ social competence.*


This study constructs a chain mediation model of the effect of maternal trait mindfulness on preschoolers’ social competence by promoting protective factors (maternal self-control) and diminishing risk factors (maternal problematic social media use). It aims to establish an empirical foundation for the intervention of preschoolers’ social competence and to expose more fully and thoroughly the mechanisms of the influence of maternal trait mindfulness on preschoolers’ social competence (Please see Figure 1).

## 2. Materials and Methods

### 2.1. Participants

This study employed convenient sampling to select parents of preschool children from one urban and rural combined kindergarten and one urban kindergarten in Qingdao, Shandong Province for questionnaire survey. A total of 661 questionnaires were distributed (all electronically), and 661 were returned, with a return rate of 100%. Five hundred twenty-four questionnaires were screened for mothers’ responses, 117 invalid questionnaires were excluded for obvious patterns of responses (e.g., choosing “1—very inconsistent” for all questions related to the measurement of social media use), and 407 valid questionnaires were obtained, resulting in a validity rate of 61.6%. Of these, 218 (53.6%) were mothers of boys, and 189 (46.4%) were mothers of girls. Mothers of students in the junior class numbered 130 (31.9%, children ages 3.5 to 4.5), those in the middle class numbered 113 (27.8%, children ages 4.5 to 5.5), and those in the senior class numbered 164 (40.3%, children ages 5.5 to 6.5). Eighty one-child households (19.7%) and 327 non-one-child families (80.3%) were included. One hundred sixty-five (40.5%) mothers only completed junior high school, 87 (21.4%) mothers completed high school or junior college, 44 (10.8%) mothers completed college, 87 (21.4%) mothers earned a bachelor’s degree, 22 (5.4%) mothers earned a master’s degree, and two (0.5%) mothers earned a Ph.D. degree.

### 2.2. Measures

#### 2.2.1. Demographic Information

The demographic information sheet that was part of the questionnaire asked about the participants’ education level, occupation, monthly household income, and the sexes and classes of their children, among other details (Please see Table 1).

To more accurately examine the relationship between maternal trait mindfulness and preschoolers’ social competence, this study followed the approach of previous published research [30] by controlling for relevant variables related to mothers and children. These variables include the child’s gender (1 = boy, 2 = girl), grade level (1 = junior class, 2 = middle class, 3 = senior class), and whether the child is an only child (1 = yes, 2 = no). We also controlled for variables pertaining to the mothers, including their education level (1 = junior high school and below, 2 = senior high school, 3 = junior college, 4 = bachelor’s degree, 5 = master’s degree, 6 = doctoral degree) and occupation (1 = company administrator, 2 = company employee, 3 = professional, 4 = service provider, 5 = worker, 6 = self-employed, 7 = unemployed, 8 = other). Additionally, we controlled for the monthly household income (1 = below 5000, 2 = 5000–10,000, 3 = 10,000–15,000, 4 = 15,000–20,000, 5 = more than 20,000).

#### 2.2.2. Maternal Trait Mindfulness

This study examined maternal levels of trait mindfulness using the Mindful Attention Awareness Scale, which was created by Mackillop and Anderson (2007) and revised by Chen et al. (2012) [16,34]. The revised Chinese version of Chen et al.’s instrument exhibits strong to its original English counterpart, thus demonstrating its validity as a reliable measurement tool in the context of mindfulness research within Chinese populations. The scale has 15 items, and subjects are asked to rank them from 1 to 6 on a scale of 1 to 6 (where 1 equals “almost always” and 6 equals “seldom”) based on which one best matches their needs for the past week (including that day). Higher scores on the unidimensional scale correspond to higher levels of trait mindfulness in the mother’s day-to-day activities. The scale’s Cronbach’s alpha coefficient for this measurement was 0.90.

#### 2.2.3. Maternal Self-Control

The Brief Self-Control Scale was created by Tangney et al. (2004), refined by Morean et al. (2014), and revised by Luo et al. (2021) in a Chinese cultural setting to assess maternal levels of self-control [18,35,36]. The revised scale not only greatly reduced the response time but also better adapted to the cultural context of the study participants, as well as having good reliability and validity. The scale consists of seven items, including two dimensions of self-regulation (1.3.5) and impulse control (2.4.6.7), of which the impulse control dimension is reverse scored. Stronger scores in both categories suggest stronger degrees of self-control. The measure was assessed on a five-point scale, with 1 being “very inconsistent” and 5 being “very consistent”. The alpha coefficients for the self-regulation and impulse control dimensions were 0.61 and 0.71, respectively, and Cronbach’s alpha coefficient for this measure was 0.63.

#### 2.2.4. Problematic Social Media Use by Mothers

The Facebook Addiction Scale (FAS), created by Koc and Gulyagci (2013) and updated by Chen et al. (2018), was used in this study to gauge maternal usage of problematic social media [37,38]. The scale was translated and backtranslated by a number of undergraduate graduate students majoring in psychology and English, and the Chinese version was determined to have good reliability and validity. Eight items on a five-point scale, ranging from 1 for “very inconsistent” to 5 for “very consistent”, are included in the scale. Higher scores on the unidimensional scale indicate mothers who use social media more problematically. This measurement’s Cronbach’s alpha coefficient was 0.82.

#### 2.2.5. Preschoolers’ Social Competence

The Preschoolers’ Social Competence and Behavior Evaluation Scale, created by LaFreniere and Jean (1996) and updated by Liu et al. (2012), was used in this study to rate social competence [39,40]. The 30-item scale uses a six-point scale from 1 to 6, ranging from 1 “never” to 6 “always” and is designed to measure social competence, emotion management, emotional expression, and difficult adjustment in children between the ages of 30 and 78 months. The revised scale by Liu et al. has good reliability and validity and is divided into three dimensions: anxiety and shyness (implicit behavior problems), anger and aggression (external behavior problems), and sensitivity and cooperation (social competence) [17]. In this study, social competence was assessed using the sensitivity and cooperation (social competence) dimensions. The current measurement’s Cronbach’s alpha coefficient for this dimension was 0.89.

### 2.3. Data Analysis

The data in this study were organized and analyzed by SPSS 22.0. First, a test for common method bias was conducted, then descriptive statistical analysis, correlation analysis, and chain-mediated effects testing using Hayes’ PROCESS macro (Model 6), all of which were done while accounting for covariates present during the investigation.

## 3. Results

### 3.1. Common Method Bias Testing

Cross-sectional surveys were used to collect the data for this study, which means that common method bias may have been present. As a result, common method bias was first identified and controlled for before data analysis.

First, the data were adjusted throughout the data collection process by employing anonymous completion, creating questions with reverse scoring, and altering how the scale’s points and values were displayed. Second, the Harman one-way test was employed to examine the results of the exploratory factor analysis for the 44-question items [41]. The analysis revealed the presence of 10 factors with eigenvalues exceeding 1. Notably, the first factor accounted for only 21.76% of the total variance, falling short of the critical criterion of 40%. This outcome suggests that the occurrence of significant common method bias in the data used for this study is unlikely.

### 3.2. Preliminary Analysis

Table 2 displays each study variable’s mean, standard deviation, and correlation matrix. The results of the correlation study revealed that the use of social media, self-control, maternal trait mindfulness, and social competence were all significantly correlated. Among them, maternal trait mindfulness was significantly positively correlated with self-control and its sub-dimensions, social competence, and significantly negatively correlated with social media use; self-control and its sub-dimensions were significantly negatively correlated with social media use and significantly positively correlated with social competence; and social media use was significantly negatively correlated with social competence.

### 3.3. Multiple Mediating Model Analysis

Based on the above results, the mediation model was tested using SPSS macro PROCESS (Model 6) with 5000 bootstrap samples, and the results are shown in Table 3 and Figure 2. The findings from the regression analysis revealed that, even after controlling for various demographic variables such as gender, grade, child status, mother’s education and occupation, and monthly household income, the mother’s level of maternal trait mindfulness exhibited a significant and positive association with her self-control (a_1_) and her child’s social competence (c) while demonstrating a significant and negative relationship with her own social media use (a_2_). Furthermore, maternal self-control exhibited a significant negative link with social media use (d) and a positive correlation with social competence (b_1_). In contrast, both the maternal level of maternal trait mindfulness (c’) and social media use (b_2_) demonstrated a positive influence on social competence to some extent.

The mediated effects analysis, as presented in Table 4, revealed a significant mediation effect of self-control and social media use in the association between maternal mindfulness and social competence (total indirect effect = 0.083, SE = 0.023, bootstrap 95% CI: [0.043, 0.133]). This mediated effect accounted for approximately 50 of the total effect (0.166). Additionally, the direct effect path (c’) between maternal mindfulness and social competence was found to be significant (direct effect = 0.083, SE = 0.035, bootstrap 95% CI: [0.019, 0.014]). This suggests that self-control and social media use partially mediated the impact of maternal mindfulness on children’s social competence. The mediating effect was composed of three pathways: the independent mediating effect of self-control (a_1_*b_1_, mediating effect = 0.052, SE = 0.019, bootstrap 95% CI: [0.016, 0.092]), which accounted for 31.32 of the total effect; the independent mediating effect of social media use (a_2_*b_2_, mediating effect = 0.019, SE = 0.011, bootstrap 95% CI: [0.004, 0.045]), which accounted for 11.45 of the total effect; and the chain-mediating effect of self-control and social media use (a_1_*d*b_2_, mediating effect = 0.012, SE = 0.005, bootstrap 95% CI: [0.004, 0.023]), representing 7.23 of the total effect. All regression coefficients are standardized.

## 4. Discussion

### 4.1. Effects of Maternal Trait Mindfulness on Preschool Child Social Competence

The findings of this study confirm Hypothesis 1, which states that maternal trait mindfulness is positively correlated with preschool child social competence. On the one hand, maternal trait mindfulness levels reflect their capacity to perceive their own experiences and feelings as well as their capacity to regulate their emotions [42]. Accordingly, mothers with higher levels of mindfulness are more likely to foster a positive home environment and harmonious parent–child relationships, increasing the likelihood that preschoolers will develop social competence through positive interpersonal relationships. However, others with higher levels of mindfulness are more mentally capable [43], which enables them to subtly filter out what many mothers with lower levels of mindfulness see as “problem behaviors in preschoolers” and exhibit a higher level of acceptance of their children. High levels of maternal mindfulness improve parenting, which in turn has a positive effect on preschoolers’ social competence [44]. Research has shown that high-quality parenting can have a positive effect on social competence, even in the face of difficult preschoolers [45]. This study extends the results of existing research and provides new empirical evidence on the effects of maternal trait mindfulness on preschoolers’ social competence.

### 4.2. The Chain-Mediating Role of Maternal Self-Control and Problematic Social Media Use in the Relationship between Maternal Trait Mindfulness and Preschoolers’ Social Competence

The present study also found that maternal self-control and problematic social media use independently mediated the relationship between maternal mindfulness and preschoolers’ social competence, and the results validated Hypotheses 2 and 3.

During childrearing, maternal self-control plays a crucial role in the development of preschoolers’ social competence. Mothers have a gender advantage in interpersonal relations since they are the major caretakers for preschoolers. As a result, they are more likely to serve as role models and take the lead in their children’s social competence development [46]. However, this benefit simultaneously puts more pressure on and demands more energy from women’s parenting. According to research, maternal ego depletion can reduce their self-control resources and lead to unbalances in their children’s impulse and control systems [47,48]. It is vital to remember that this imbalance can further seep into the mother’s parenting process and negatively affect the preschool child’s social competence [49]. According to Bialy (2006) [50], the mother will be better able to tolerate and accept these strong emotions if she has a higher level of mindfulness. She will also be better able to manage the self-control imbalance, reduce inappropriate reactions, and reduce behaviors in parent–child interactions, creating a secure interpersonal setting for the social competence development of preschoolers.

With the increasing popularity of online social media, problematic social media use by mothers has become a risk factor for social competence in preschoolers. On the one hand, the portability and “fragmented” use of smartphones make problematic social media use highly automated [7] and maternal frequent chatting, entertainment, and hanging out on social media take up a lot of time for parent–child interaction, causing mothers and children to miss more meaningful real-world experiences and delaying the opportunity for children to practice their social competence [51]. However, mothers who are addicted to social media will frequently access the “wonderful experiences” of others [52], which may lead to upward social comparison [53], comparing their own and their children’s current situation with the “glamorous” lives of others on social media, resulting in psychological loss and anxiety [54], which can affect the quality of parent–child interactions.

Fortunately, a maternal trait of mindfulness serves as a protective element and can aid them in developing clarity of purpose, effective control over their conduct [55], and successful mitigation of their automatic social media use. Additionally, the “accepting attitude” encouraged by mindfulness might mitigate the anxiety brought on by upward social comparisons and shield preschoolers from the damaging impacts of their maternal negative emotions and behaviors.

Additionally, our research revealed that maternal self-control and questionable social media use play a chain-mediating role between maternal trait mindfulness and preschoolers’ social competence, supporting Hypothesis 4. The development of preschoolers’ social competence is positively impacted by maternal mindfulness because it increases self-control, which in turn serves to minimize the negative consequences of problematic social media use by mothers. The chain-mediated results validate the main idea of attachment theory that there is a strong association between healthy parent–child attachment and preschoolers’ social competence and emotion regulation, and from this perspective, many factors such as maternal traits (e.g., trait mindfulness) and self-control and problematic social media use behaviors can profoundly affect the quality of parent–child attachment directly or indirectly [56,57].

The significant findings of the chain-mediation model reveal how maternal traits of mindfulness, self-control, and problematic social media use combine to influence the social competence of preschoolers. Our research findings demonstrate that maternal self-control and problematic social media use not only independently mediate the impact of maternal mindfulness on the social competence of preschoolers but also work together in a chain fashion. This implies that maternal mindfulness first affects self-control, which in turn influences problematic social media usage behavior, ultimately impacting the social competence of preschoolers. This finding contributes to a better understanding of how maternal mindfulness influences the competence of preschoolers and provides a theoretical basis for designing relevant intervention measures.

In summary, the unique contribution of our study lies in filling the literature gap on how maternal traits of mindfulness affect the social competence of preschoolers. While previous research has focused on the impact of maternal mindfulness on the development of social competence in preschoolers, the mediating mechanisms have not been thoroughly understood. Our study offers a new perspective by exploring the mediating effects of maternal self-control and problematic social media use, revealing how maternal mindfulness influences the social competence of preschoolers.

## 5. Limitations and Directions for Future Research

First off, following the study’s objectives, only data from maternal accounts were utilized in the study, and the findings did not adequately account for significant others in the home and garden that affect preschoolers’ development of social competence. On the one hand, preschoolers’ social competence can also be greatly influenced by their fathers at home. Future research will focus on how fathers and mothers differ from one another in terms of gender and social roles, as well as how they prefer to utilize the internet and exhibit trait mindfulness. However, due to disparities between the home and school environments, children’s social competence may be displayed differently in the school setting and is also influenced by teachers and peers. To strengthen the trustworthiness of the data and try to prevent potential bias in single-subject reporting, future research should utilize a multi-subject reporting technique to gather information on children’s social competence.

Second, because this study is cross-sectional, future research can use longitudinal tracking to examine the development trend of preschoolers’ social competence. This will allow researchers to determine whether the rate of development of preschoolers’ social competence is constant over time as well as how differentiating personal characteristics and environmental factors interact to affect this development. This will help develop strategies for focusing on the social competence of preschoolers.

Once more, the sociocultural milieu in which preschoolers live plays a critical role in the development of their social competence. The social competence of preschoolers was explored in this study within a Chinese cultural framework. However, gender variations in social competence were not specifically examined. Girls are frequently expected in Chinese culture to behave more submissively and adhere to their parents’ and instructors’ instructions, whereas boys are encouraged to be forceful and persistent. To further the understanding of preschoolers’ social competence, future research should concentrate on examining preschoolers’ social development in “specific” sociocultural situations. It is clear that sociocultural expectations have a subtle influence on preschoolers’ social competence.

Finally, the cross-sectional design used in this study may limit our ability to accurately disentangle the causal relationships between maternal problematic social media use and child social competence. Specifically, we cannot exclude the possibility of reverse causality, where low social competence in children may lead to more frequent social media use in mothers. Furthermore, due to limitations in sample size and representativeness, our findings may not be universally applicable. Although this study revealed the correlation between maternal problematic social media use and child social competence, future studies should use longitudinal designs and more representative samples to continue exploring this issue for more precise causal inference. We also recommend that future researchers employ a multi-method approach to gather data for more comprehensive insights.

## 6. Practical Implications

Despite these limitations, our research results have significant practical implications. Firstly, mothers should try to minimize their use of mobile phones, especially social media, when interacting with their children. Additionally, mothers should consciously enhance their self-control, which can not only have a positive impact on the social development of preschool children, but also buffer the risks posed by problematic social media use. In fact, stable and positive mother–child relationships are the foundation of the social development of preschool children. Our study confirms the positive impact of maternal trait mindfulness on the social development of preschool children and identifies the chain-mediating effects of maternal self-control and problematic social media use in this relationship. The results of this study will provide new ideas for the intervention and improvement of social abilities in preschool children.

## Figures and Tables

**Figure 1 behavsci-13-00805-f001:**
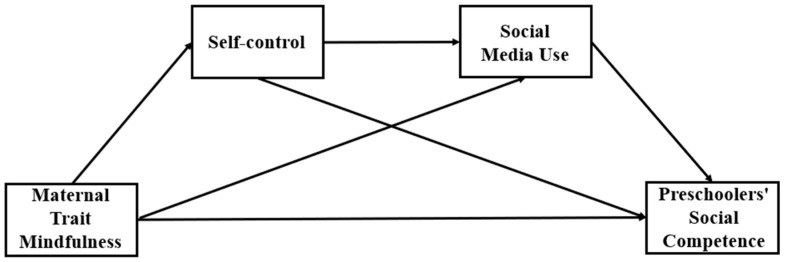
Hypothetical Model of the Study.

**Figure 2 behavsci-13-00805-f002:**
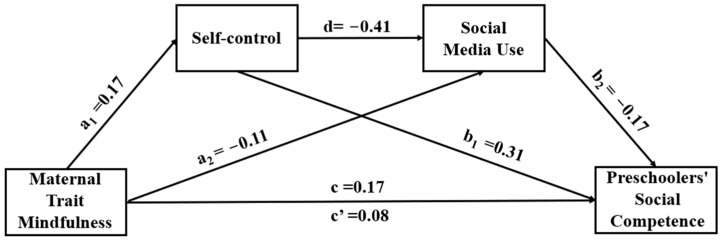
The pathway of the mediating model.

**Table 1 behavsci-13-00805-t001:** Demographic characteristics of the participants (N = 407).

Characteristics (Code)	n (%)
Child’s gender	
Boy	218 (53.6)
Girl	189 (46.4)
Child’s grade	
Junior Class of kindergarten	130 (31.9)
Middle Class of kindergarten	113 (27.8)
Senior Class of kindergarten	164 (40.3)
Only child	
Yes	80 (19.7)
No	327 (80.3)
Education level of the child’s mother	
Junior high school and below	165 (40.5)
Senior high school	87 (21.4)
Junior college	44 (10.8)
Bachelor’s degree	87 (21.4)
Master’s degree	22 (5.4)
Doctoral degree	2 (0.5)
Mother’s occupation	
The company’s administrator	9 (2.2)
The company’s employee	50 (12.2)
The professional	48 (11.8)
The service provider	23 (5.7)
The worker	90 (22.1)
The self-employed	54 (13.3)
The unemployed	87 (21.4)
Other	46 (11.3)
Average monthly family income (CNY)	
Below 5000	162 (39.8)
~5000–10,000	120 (29.5)
~10,000–15,000	44 (10.8)
~15,000–20,000	30 (7.4)
More than 20,000	51 (12.5)

**Table 2 behavsci-13-00805-t002:** Descriptive statistics and correlation analysis.

	M	SD	1	2	3	4	5
1 Maternal mindfulness	74.03	11.46	1				
2 Self-discipline	11.38	2.19	0.50 ***	1			
3 Impulse control	14.27	3.60	0.23 ***	0.11 *	1		
4 Total self-control score	25.65	4.41	0.44 ***	0.59 ***	0.87 ***	1	
5 Social media use	19.05	6.36	−0.32 ***	−0.18 ***	−0.35 ***	−0.37 ***	1
6 Social competence	41.60	7.65	0.25 ***	0.31 ***	0.16 **	0.29 ***	−0.25 ***

Note: *** *p* < 0.001, ** *p* < 0.01, * *p* < 0.05.

**Table 3 behavsci-13-00805-t003:** Multiple regression analysis.

Regression Model		Regression Coefficient Significance				Overall Fitting Index		
Dependent variable	Independent variable	β	t	LLCI	ULCI	R	R2	F
Social competence	Gender	0.06	1.36	−0.44	2.38	0.38	0.14	9.55 ***
	Grade	0.22	4.63 ***	1.15	2.85			
	Only child	−0.01	−0.25	−2.10	1.62			
	Mother’s education level	0.04	0.62	−0.54	1.03			
	Mother’s occupation	−0.02	−0.28	−0.46	0.34			
	Average monthly family income	0.16	2.57	0.21	1.61			
	Maternal mindfulness	0.27	5.77 ***	0.12	0.24			
Self-control	Gender	0.05	1.21	−0.30	1.25	0.47	0.22	16.44 ***
	Grade	0.13	2.87 **	0.21	1.15			
	Only child	−0.07	−1.43	−1.76	0.28			
	Mother’s education level	−0.07	−1.03	−0.66	0.20			
	Mother’s occupation	0.06	1.19	−0.09	0.35			
	Average monthly family income	0.04	0.68	−0.25	0.52			
	Maternal mindfulness	0.44	10.02 ***	0.14	0.20			
Social media use	Gender	0.02	0.42	−0.90	1.39	0.43	0.18	11.22 ***
	Grade	−0.05	−0.97	−1.04	0.35			
	Only child	0.02	0.43	−1.19	1.85			
	Mother’s education level	0.13	1.93	−0.01	1.26			
	Mother’s occupation	−0.03	−0.61	−0.43	0.22			
	Average monthly family income	−0.12	−1.97	−1.14	−0.001			
	Maternal mindfulness	−0.21	−4.15 ***	−0.17	−0.06			
	Self-control	−0.26	−5.12 ***	−0.53	−0.23			
Social competence	Gender	0.06	1.21	−0.53	2.23	0.43	0.19	10.22 ***
	Grade	0.19	4.02 ***	0.87	2.55			
	Only child	0.004	0.07	−1.75	1.89			
	Mother’s education level	0.08	1.09	−0.34	1.20			
	Mother’s occupation	−0.03	−0.59	−0.51	0.27			
	Average monthly family income	0.14	2.22 *	0.09	1.46			
	Maternal mindfulness	0.15	2.95 **	0.03	0.17			
	Self-control	0.16	2.99 **	0.09	0.46			
	Social media use	−0.14	−2.74 **	−0.28	−0.05			

Note: *** *p* < 0.001, ** *p* < 0.01, * *p* < 0.05.

**Table 4 behavsci-13-00805-t004:** Indirect effects of self-control and social media use.

	Effect	Boot SE	Boot LLCI	Boot ULCI	Ratio of Total
Total indirect effect	0.083	0.023	0.043	0.133	50%
Indirect effect 1 (a_1_*b_1_)	0.052	0.019	0.016	0.092	31.32%
Indirect effect 2 (a_2_*b_2_)	0.019	0.011	0.004	0.045	11.45%
Indirect effect 3 (a_1_*d*b_2_)	0.012	0.005	0.004	0.023	7.23%

## Data Availability

No new data were created for this study.

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
