# Peer review of "The Effect of Maternal Trait Mindfulness on Preschoolers’ Social Competence: The Chain-Mediating Role of Maternal Self-Control and Problematic Social Media Use"

_behavsci, 2023, doi:10.3390/bs13100805_

Round 1
Reviewer 1 Report
Manuscript ID: behavsci-2584497
Title: The Effect of Maternal Trait Mindfulness on Preschoolers' Social Competence: The Chain Mediating Role of Maternal Self-control and Problematic Social Media Use
Thank you for inviting me to review this manuscript. This manuscript adopts a quantitative research method to verify the chain mediating role played by Maternal Self-control and Problematic Social Media Use in the relationship between Maternal Trait Mindfulness and Preschoolers' Social Competence. The findings of this study have some theoretical and practical value for family upbringing in the digital age. However, the manuscript still has some areas that need to be revised and improved.
1. Page 2, lines 49-52, this section identifies the research objectives of the study, but there is no need for a single sentence in a paragraph. It is suggested that the sentence be suitably integrated into other paragraphs.
2. Please add a section about theoretical framework before you proposed the chain mediation model of the current study.
3. Paragraph starting from line 87 reviewed relationships between mothers’ problematic screen use and children’s social competence explored by previous studies. I think more updated and relevant studies provide a context for this article:
Li, H., Luo, W., & He, H. Association of Parental Screen Addiction with Young Children’s Screen Addiction: A Chain-Mediating Model. Int. J. Environ. Res. Public Health, 19, 12788. doi: 10.3390/ijerph191912788.
This study offers pertinent findings and methodologies that align closely with the themes of your manuscript. Incorporating insights or comparisons from this paper could provide a more comprehensive perspective and potentially strengthen the arguments you present.
4. Page 3, line 123, " In this study, parents were chosen from one urban, one rural, and one urban kindergarten in China". In this sentence, "one urban" appears twice. If the study was conducted with mothers from two urban kindergartens, it is suggested that the word "two urban" be used instead.
5. Was this study a questionnaire administered to the mothers of all children in three kindergartens? Was an electronic or paper questionnaire used? I think this information should be clearly stated.
6. In this study, demographic information such as sex and grade of the young child, mother's education level, occupation and monthly income were used as control variables. The authors should include a brief explanation of the reason for the control variables in the text.
7. In addition, I think t-tests or ANOVAs are needed to report whether you found group differences and then you can claim that you would use these variables as control variables. Also, if you would like to use control variables, you should add some previous studies regarding these differences in the literature review section.
8. In addition, mother's occupation is the obvious nominal variable, so did the authors dummy it when they included it as a control variable in the regression equation. If not, would the authors please provide further explanations as to whether this would affect the results of the study?
9. P7, Table 3. Are the regression coefficients presented in the table standardized or unstandardized regression coefficients?" β" is the symbol for a standardized regression coefficient, but the data in the table may be an unstandardized regression coefficient. Please ask the authors to check and clarify.
10. P8, Figure 1. The authors should make it clear that this figure shows standardized regression path coefficients.
In general, the current version has a lot of misspellings of words, grammatical errors, and typo. I will suggest you ask some native speakers helping you improve your English academic writing.
Reviewer 2 Report
I suggest authors include a conceptual framework and create a diagram to depict the findings.
Reviewer 3 Report
This is an interesting study maternal trait mindfulness and preschoolers' social competence. I have several comments to improve the manuscript further:
1. First, ensure consistent terminology. For instance, the term "mobile social networks" in line 40 might be more commonly understood as "social media". Similarly, the terms "kids" (line 37) and "children" are used interchangeably. For consistency and a more academic tone, it's better to use "children."
2. the term "fragmented" use of cell phones needs further explanation or a citation for clarification.
3. The conclusion wraps up the introduction well, but consider clarifying the unique contribution of the study. What gap in the existing literature is it addressing? Why is the chain mediation model of significance?
4. The clarify the number of parents initially approached or surveyed to derive the 61.6% effective rate.
5. Please clarify the term "seemingly regular responses" – this seems ambiguous and could benefit from a more precise description.
6. Please provide the age range or mean age (and perhaps other demographic details) of the mothers surveyed
7. When mentioning updates or revisions to a tool (e.g., Mindful Attention Awareness Scale revised by Chen et al. in 2012), it may be helpful to mention briefly what the revisions encompassed or why they were necessary.
8. The methodology briefly mentioned controlling for covariates but did not list or discuss what these covariates were. This is very problematic.
9. Please clarify which specific covariates were included in the regression models and their rationale
10. In the discussion section, I would like the authors to acknowledge potential reverse causation in their findings due to the cross-sectional nature of the design. For instance, lower social competence could be the antecedent rather than outcome of social media use. Reverse causation has been a big issue in the social media literature. The authors should elaborate and highlight this in their limitation. Please see the following paper that is relevant to the discussion: Does social media use increase depressive symptoms? A reverse causation perspective. (2021) Frontiers in Psychiatry, 12, 641934.
11. The study briefly touches upon gender expectations in Chinese culture but does not delve into how these differences might influence their results.
12. While the study acknowledges its Chinese cultural framework, there's a potential for cultural bias. It would be beneficial if the study could provide comparisons or insights into how the findings might differ in other cultural contexts.
Round 2
Reviewer 1 Report
Thank you for all your responses and revisions.
This version is better than the original one.
Reviewer 3 Report
The authors have addressed my comments well.